# The Accuracy of a Screening Tool in Epidemiological Studies—An Example of Exhaled Nitric Oxide in Paediatric Asthma

**DOI:** 10.3390/ijerph192214746

**Published:** 2022-11-09

**Authors:** Kamil Barański, Vivi Schlünssen

**Affiliations:** 1Department of Epidemiology, Medical University of Silesia, 40-055 Katowice, Poland; 2Research Unit for Environment, Occupation and Health, Department of Public Health, Danish Ramazzini Centre, Aarhus University, 8000 Aarhus, Denmark

**Keywords:** accuracy, screening tool, fractional exhaled nitric oxide, FeNO

## Abstract

Diagnostic tests are widely used in medicine, especially in the clinical setting. The doctor’s decision regarding the treatment process is mostly based on the result of the diagnostic test. The value of the test is expressed by its accuracy. It is easier to verify the accuracy of a diagnostic test in a clinical setting in comparison to an epidemiological setting. Moreover, a very good test may not work in epidemiological settings in the same effective way as in a clinical setting, especially because the accuracy is affected by the prevalence of the disease. The aim of the study is to assess the accuracy of FeNO measurement in different respiratory disorders or symptoms, including their prevalence, in a childhood population. The secondary aim is to suggest the optimal FeNO cut-off for epidemiological screening for respiratory diseases and symptoms. Methods: The cross-sectional study included 447 children (50.8% boys and 49.2% girls) aged 6–9 years. An adapted version of the ISAAC questionnaire was used for the assessment of the respiratory status. FeNO was measured with an electrochemical device (Niox Mino) according to ERS/ATS recommendations. For interpretation, the FeNO cut-off values of 20 parts per billion (ppb), 25 ppb and 35 ppb were applied taking the real-life prevalence of the disease or symptoms into consideration and also for simulated prevalences of 20%, 30%, 40%, 50% for the interpretation of the accuracy of the test. The accuracy was calculated according to the following formula: Accuracy = (Prevalence) (Sensitivity) + (1- Prevalence) (Specificity). The area under the curve was calculated based on logistic regression. Results: For all respiratory outcomes, FeNO accuracy decreased with increasing prevalence, and in general the area under the curve (AUC) was low. The highest FeNO accuracy was found for any asthma diagnosis (with possible coexisting diseases/symptoms), i.e., 78.6%, 92.8% and 88.5% for FeNO cut-offs >19 ppb, >24 ppb and >34 ppb, respectively. The AUC was 0.628. For the same FeNO cut-offs, the accuracy of an asthma diagnosis without any coexisting diseases and symptoms was 81.2%, 87.5%, 92.9%, respectively, with an AUC of 0.757. Conclusion: FeNO accuracy decreases with increasing prevalence of the respiratory disease and symptoms. The best accuracy for the FeNO cut-off in the screening of asthma for epidemiological purposes is 35 ppb. For isolated asthma, the best accuracy for FeNO was 92.9%.

## 1. Introduction

The diagnostic test plays a key role in confirming a diagnosis in the clinical setting, and also for controlling the effects of treatment [1]. In the epidemiological setting, the situation is different. The diagnostic tool is mostly used for screening on the population level. In this case, a range of factors influence the effectiveness of its utility. Among others, these are the prevalence, specificity (SPE), sensitivity (SEN) and positive and negative predictive value (PPV and NPV). The costs of implementation, as well as the costs of treatment, are important [2]. However, from a diagnostic point of view, only values related to probability are crucial. Other factors that are important for the assessment of the utility of a diagnostic tool are the difficulty of implementation, safety for the subject and whether it is easy to apply [3]. For respiratory diseases, which are frequent in the young population, one of the available diagnostic tools that is easy to implement is fractional exhaled nitric oxide (FeNO), a biomarker of eosinophilic inflammation in airways [4,5]. FeNO measurement is well established in the clinical setting and it is used in monitoring and controlling the management of asthma [6]. It is less known how FeNO measurement acts in epidemiological conditions. The aim of the study is to analyse the accuracy of FeNO in epidemiological settings, including differences in the prevalence of respiratory symptoms reported in a questionnaire.

## 2. Materials and Methods

The data collection covered 2017–2020. The participants were 447 school-aged children from primary schools in 4 cities (Bytom, Chorzów, Tychy and Zabrze) located in the Silesian Voivodship (Poland). The modified version of the Study of Asthma and Allergies in Childhood (ISAAC) questionnaire [7] was used to assess the respiratory status of the children. The following respiratory/allergic outcomes were included in the analysis: asthma (ever diagnosed by a physician), asthma without coexisting disorders, cough at night, cough during the day, allergic rhinitis, dyspnoea during the day, dyspnoea during the night, diagnosed by a physician in the past or reported by legal guardian or parents.

FeNO measurement (electrochemical NIOX MINO device, Circassia, Stockholm, Sweden) was performed according to ERS/ATS recommendations [8]. The test was applied by a trained and certified researcher. FeNO was measured in the middle of the week (Tuesday, Wednesday or Thursday) to decrease tobacco smoke exposure if parents declared a positive smoking status. Children were asked to not drink water before the FeNO measurement, avoid physical activity and avoid food, which could influence the FeNO levels.

The accuracy was calculated with the following formula [9]:Accuracy = (Prevalence) (Sensitivity) + (1-Prevalence) (Specificity)

The accuracy of FeNO was calculated with the real-life prevalence of the disease/symptom generated through the questionnaire and through the simulated prevalence of the disease/symptom, 20%, 30%, 40% and 50%.

For each specific respiratory/allergic outcome, the frequency of true positive (TP) and true negative cases (TN) and false positive (FP) and false negative (FN) cases was calculated in relation to three defined thresholds of FeNO values (20, 25, 35 ppb). The set of variables used to assess the diagnostic accuracy of FeNO measurement included sensitivity (SENS = TP/(TP + FN), specificity (SPEC = TN/(TN + FP) and positive and negative predictive value (PPV = TP/(TP + FP); NPV = TN/(TN + FN) [10]. The area under the curve (AUC) was calculated to present FeNO accuracy as a continuous variable [11] for each diagnosis/symptom.

All analyses were performed using the SAS statistical package (SAS Institute Inc., Cary, NC, USA, version 9.4).

## 3. Results

In the study, there were 37/447 cases (8.2%) of any asthma diagnosed by a physician. For FeNO, the best SEN of 32.4% was noticed for FeNO cut-off >19 ppb, the best SPE was 98.5% for cut off > 24ppb, the best PPV was 24.1% for FeNO cut-off > 34 ppb and the best NPV was 93.1% for all FeNO cut-offs. When considering asthma without any coexisting disorders, such as wheezing in the chest, cough, dyspnoea or allergic rhinitis, the best SEN of 40% was found at FeNO cut-off >19 ppb, the best SPE was 93.6% for FeNO cut-off > 34 ppb, the best PPV was 3.4% for FeNO cut-off > 34 ppb and the best NPV of 99.1% was noticed for FeNO cut-offs > 19 ppb and 24 ppb. When considering different types of cough (during the night or during the day), the best SEN (19.5%) was found for cough during the day for FeNO > 19 ppb, the best SPE (95.7%) was found for FeNO >34ppb, the best PPV was 68.9% for FeNO > 34 and the best NPV was 53.5% for FeNO > 19 ppb. For allergic rhinitis, the best SEN of 15.5% was noticed for FeNO cut-off > 19 ppb, the best SPE was 84.9 for FeNO cut-off > 34 ppb, the best PPV was 63.8% for FeNO cut-off > 19 ppb, and the best NPV was 21.5% for FeNO cut-off > 34 ppb. When considering symptoms of dyspnoea during the day or night, the best SEN of 29.4% was found for FeNO cut-off > 19 ppb, the best SPE was 93.9% for FeNO cut-off 34 ppb, the best PPV was 10.3% for FeNO cut-off > 34 ppb and the best NPV was 97.4% for FeNO cut-off > 24 ppb.

All results of sensitivity, specificity, positive predictive value and negative predictive value are shown in Table 1.

The diagnostic accuracy was presented as the area under the curve (AUC), with real prevalence (result from questionnaire screening) and simulated prevalence, at 20%, 30%, 40% and 50%. For any asthma (8.2% from the questionnaire), the best accuracy was for FeNO cut-off 24 ppb, and the area under the curve was 0.628. Higher accuracy was seen for asthma without any coexisting disorders; the prevalence was 1%, the best accuracy was in children with FeNO > 34 ppb, and AUC = 0.752. For cough during the day or during the night, the best accuracy value was noted in the simulated prevalence of 20%, increasing with increasing FeNO cut-off. Similar results were seen for allergic rhinitis. For dyspnoea during the day or night, the best accuracy results were reported within the real prevalence of the symptoms and increasing with increasing FeNO cut-off. The AUC was low and ranged between 0.520 and 0.576; see Table 2.

The best FeNO accuracy (92.8%) for isolated asthma was shown for the highest FeNO cut-off (>34 ppb) and with the lowest simulated prevalence of asthma (1%). The accuracy of FeNO was acceptable with asthma prevalence ranging from 1% to 10%; for FeNO cut-off > 19, the accuracy was 81.1–77.4%; for FeNO cut-off > 24 ppb, it was 80.9–73.4%, and for the highest cut-off > 34 ppb, it was 92.8–86.2; see Figure 1.

## 4. Discussion

In our study, the most valuable results of accuracy were found in relation to any asthma or asthma without coexisting diseases/symptoms. For the first mentioned disorder and its real prevalence, the accuracy varied from 78.6% to 92.8% with an AUC of 0.628. For the asthma diagnosis without any coexisting disorders or symptoms, the accuracy varied from 81.2% to 92.9% and the AUC was 0.757. For dyspnoea, FeNO showed an acceptable accuracy but AUCs were 0.576 and 0.511, which are interpreted as noninterpretative results [12]. The accuracy for other diseases such as allergic rhinitis or symptoms such as cough varied from 49.4% to 66.2%. Interpretation of the accuracy of diagnostic or screening tools is challenging and depends on different factors. Considering the area under the curve based on logistic regression, where the independent variable is continuous, an AUC value equal to 0.5 means that the test has no discriminating power. If the AUC value is higher than 0.5, the diagnostic tool has discriminating ability. However, it is still difficult to determine which level of AUC is sufficient and can be taken as a minimum satisfactory result [12]. In our study, the diagnosis of isolated asthma in relation to the highest recommended cut-off in children (cut-off > 35ppb) showed AUC = 0.757, which means that FeNO at 75% is able to detect abnormal results in comparison to children without any respiratory symptoms.

The accuracy, which includes also the prevalence of the disease (in our study, isolated asthma was seen among 1% of the children), was 92.8% for FeNO cut-off > 34ppb. The accuracy decreased with decreasing FeNO cut-off levels (which influenced sensitivity and specificity), as well as with the increasing prevalence of the disease/symptom.

Our study demonstrated that sensitivity and specificity vary according to the prevalence of the disease and symptoms when using FeNO as a diagnostic tool. Moreover, the accuracy of FeNO also changes across different prevalences and with the chosen cut-off value. The calculated values of specificity and sensitivity in our study for any asthma diagnosed by a physician or any isolated asthma correspond with values expressed in a systematic review by Karrasch et al. [4]. At the same time, the results of AUC for FeNo in relation to asthma also correspond with findings presented by other authors. In Kellerer’s study, the AUC ranged from 0.672 to 0.776 [13].

Prevalence is normally a stable indicator in relation to chronic diseases in the population. However, the situation of asthma in Poland is different; the prevalence of asthma reported across a 21-year period increased from 3.4% in 1993 to 12.6% in 2014 [14]. However, our simulation suggests that for increasing prevalence, FeNO’s accuracy will decrease, and this may pose a challenge for the future use of FeNO for epidemiological purposes.

Improvement of the diagnostic accuracy of a diagnostic tool is not an easy task, and there are standards for the reporting of diagnostic accuracy studies; however, a universal protocol is still lacking [15]. It seems that combining outcomes might increase the accuracy [16]; however, it is difficult (due to costs and time restraints) to implement multiple diagnostic tools as a screening method for one disease at the population level.

The accuracy of a diagnostic tool is crucial. For FeNO, it seems that the specificity are higher than the sensitivity, which suggests that FeNO measurement is more useful for confirming rather than ruling out the disease [4].

### Limitations

Our analysis has some advantages and limitations. We worked with real data. Moreover, we analysed a fair number of participants in comparison to other studies. On the other hand, we are aware that our study is based on the declarative reports of the legal guardians/parents of children. The prevalence of asthma in our study was relatively low but corresponded with the reported prevalence of the disease in Poland. At the same time, it should not affect the threshold by prevalence. Moreover, we did not analyse FeNO in relation to the use of corticosteroids in asthma treatment, but this is a problem in epidemiological studies, since parents may not be aware of the type of used drugs in the current treatment of their children. Such problems were found not only in observational studies [17]. We did not control for possible confounding factors; however, the linkage was between FeNO measurement and the outcome expressed as a disease or symptom. For instance, sex and age did not play a crucial role. Moreover, the sex, as well as age, did not differ between the groups. The characteristics of the children were expressed elsewhere [18].

## 5. Conclusions

FeNO accuracy decreases with the increasing prevalence of the respiratory disease and symptoms. However, the accuracy remains relatively stable if the prevalence does not exceed 10%. The best accuracy in the screening of asthma for epidemiological purposes is achieved with a cut-off of 35 ppb. The accuracy of FeNO when including the prevalence of the disease results in higher scores in comparison to accuracy expressed as the area under the curve. This might be because of the incorrectly reported occurrence of asthma or other symptoms involving the respiratory system in the questionnaire.

## Figures and Tables

**Figure 1 ijerph-19-14746-f001:**
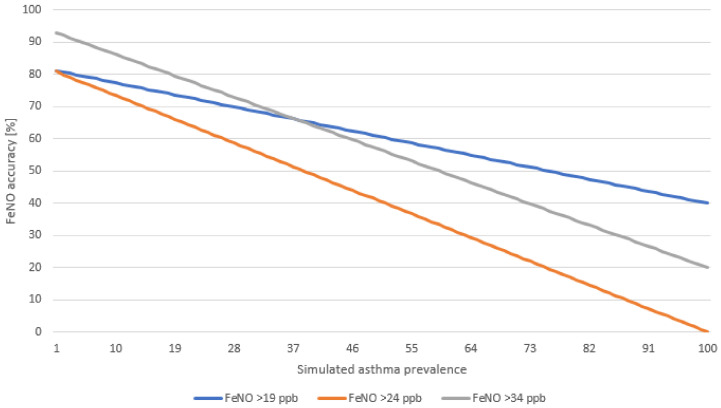
FeNO accuracy in epidemiological asthma diagnosis presented with the simulated prevalence of asthma and three FeNO cut-offs.

**Table 1 ijerph-19-14746-t001:** Diagnostic accuracy (sensitivity, specificity, positive and negative predictive value) of FeNO in detecting treated asthma, allergic rhinitis, cough at night and day, dyspnoea at night and day in children vs. children without any respiratory symptoms.

Outcome	FeNO [ppb]	SEN %	SPE %	PPV %	NPV %
Any asthma n = 37	>19	32.4	82.6	14.4	93.1
	>24	27.0	98.5	18.8	93.1
	>34	18.9	94.6	24.1	93.1
Asthma without coexisting disorders n = 5	>19	40.0	81.6	2.4	99.1
	>24	20.0	88.2	1.8	99.1
	>34	20.0	93.6	3.4	98.9
Cough during the night n = 234	>19	19.2	82.1	54.2	48.0
	>24	13.6	90.0	60.3	48.7
	>34	8.5	95.7	68.9	48.8
Cough during the day n = 210	>19	19.5	82.2	49.4	53.5
	>24	14.2	90.3	56.6	54.3
	>34	7.1	94.0	51.7	53.3
Allergic rhinitis n = 106	>19	15.5	71.7	63.8	20.8
	>24	8.8	78.3	56.6	21.0
	>34	3.8	84.9	44.8	21.5
Dyspnoea during the night n = 17	>19	29.4	81.8	6.0	96.7
	>24	23.5	88.6	7.5	96.7
	>34	17.6	93.9	10.3	96.6
Dyspnoea during the day n = 12	>19	16.6	81.3	3.7	97.2
	>24	16.6	88.2	3.7	97.4
	>34	8.3	93.5	3.4	97.3

**Table 2 ijerph-19-14746-t002:** Diagnostic accuracy expressed as area under the curve, accuracy with a known prevalence of disorder and accuracy with real (R) and simulated (S) prevalence of the disease 20%, 30%, 40%, 50%.

Outcome	FeNO [ppb]	Area under the Curve	Accuracy Real Prevalence	Accuracy S.20%	Accuracy S.30%	Accuracy S.40%	Accuracy S.50%
Any asthma n = 37	>19	0.628	78.6	72.6	67.5	62.5	57.5
	>24		92.8	84.2	77.1	69.9	62.8
	>34		88.5	79.5	71.9	64.3	56.8
Asthma without coexisting disorders n = 5	>19	0.757	81.2	73.3	69.1	65.0	60.8
	>24		87.5	74.6	67.7	60.9	54.1
	>34		92.9	78.9	71.5	64.2	56.8
Cough during the night n = 234	>19	0.520	49.4	69.5	63.2	56.9	50.7
	>24		50.3	74.7	67.1	59.4	51.8
	>34		50.4	78.3	69.5	60.8	52.1
Cough during the day n=210	>19	0.521	53.4	69.7	63.4	57.1	50.9
	>24		55.3	75.1	67.5	59.9	52.3
	>34		54.0	76.6	67.9	59.2	50.6
Allergic rhinitis n = 106	>19	0.570	58.8	60.5	54.8	49.2	43.6
	>24		62.3	64.4	57.5	50.5	43.6
	>34		66.2	68.7	60.6	52.5	44.4
Dyspnoea during the night n = 17	>19	0.576	80.2	71.3	66.1	60.8	55.6
	>24		86.6	75.6	69.1	62.6	56.1
	>34		91.6	78.6	71.0	63.4	55.8
Dyspnoea during the day n = 12	>19	0.511	80.0	68.4	61.9	55.4	49.0
	>24		86.8	73.9	66.7	59.6	52.4
	>34		91.8	76.5	67.9	59.4	50.9

## Data Availability

The data are available on a reasonable request from the corresponding author.

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
