# Peer review of "The Accuracy of a Screening Tool in Epidemiological Studies—An Example of Exhaled Nitric Oxide in Paediatric Asthma"

_ijerph, 2022, doi:10.3390/ijerph192214746_

Round 1

Reviewer 1 Report

I appreciate the opportunity to review the manuscript, “The accuracy of a screening tool in epidemiological studies –An example of exhaled nitric oxide in paediatric asthma”. This study by BaraÅ„ski et al. utilized FeNO measures from 447 children to assess their respiratory status and outcomes. The authors identified that FeNO accuracy decreased with increasing disease prevalence or symptoms and selected 35 ppb as the best FeNO cut off in the screening of asthma for epidemiological purposes. Overall, this is a well written concise paper, however I have some suggestions for improvement:

Abstract

1.    Line 18: Are you missing the numbers for % of boys and % of girls?

2.    Line 20: Use full form of ppb – parts per billion at the first occurrence, Abbreviation later is okay

3.    Line 27: An extra ppb is there – 24 ppb> ppb

4.    Lines 26-31: What was the best accuracy for the FeNO cut off of 35 ppb? Mention it again in the last sentence of the abstract. Also, it is not clear whether this cut off is for asthma diagnosis without any other co-existing symptoms. The best accuracy for asthma diagnosis is 92.8% but the FeNO threshold for that does not match to 35 ppb so this is confusing paragraph. And where you mention the accuracy of 81.2%, 87.5%, 92.9%, are these for the same FeNO cut off as in line 26-27? If yes, then start with something like: “For the same FeNO cut offs, the accuracy of …”. Please make this section of results clearer.

5.    It would make it more significant and appealing to the reader if the authors could highlight the broader perspective in the conclusion how this could be useful in epidemiological setting for eg. The last sentence of discussion and conclusion in the end touch upon this but the abstract does not highlight this at all. Also, this threshold of 35 ppb is often already used as a marker of airway inflammation in children. So, what is novel here? Why not trust what is clinically already known. Can you comment?

Methods: If this is real data, why is consent or assent not applicable here?

Results:

1.    The FeNO threshold/cut off is based on 8.2% of asthma cases. This is small sample size to test your hypothesis and accuracy for asthma diagnosis. You have better sample sizes for cough or allergic rhinitis, but FeNO likely is not the best diagnostic/discriminatory tool for those with low AUC. Can authors comment on how this threshold is representative of asthma diagnosis based on such a small number? This would be crucial as the whole point of your conclusion is basically based on 8.2% of asthma cases.

2.    Can you find another asthma population to replicate your findings for the purpose of generalizability as well as with better sample sizes? There are so many publicly available datasets too.

3.    Are there any confounding factors that could affect your findings like do you see differences by sex? You may end up with further small samples too. Would AUC change by these confounding factors? Would be useful to say this in methods and have a table of any demographics for the population you have for results may be.

Discussion, Line 130: Correct the sentence as: “The accuracy for other diseases like allergic rhinits…. varied from….49.4% to 66.2%.”

Minor comments: Check the use of full stops, commas throughout the manuscript

Results, Line 83: There is a double c in “FeNO ccut-offs” and full stop is missing after.

Author Response

Dear Reviewer,

thank you for your time and effort that you put to improve our manuscript. Please, kindly see the attached responses.

Kind regards,

Reviewer 2 Report

The aim of this manuscript is twofold: (1) evaluate accuracy of FeNO in detecting asthma or other respiratory symptoms and (2) suggest an optimal FeNO cutoff using cross-sectional data on ~450 children ages 6-9. The issue of FeNO as a diagnostic tool for asthma has been well discussed in the literature and is quite complex. The authors do the issues poor justice here. They need to read more deeply on the subject. Indeed, in the early phases of the study of FeNO there was hope that it would be useful in diagnosing incident asthma but now we understand that FeNO is much more useful to help guide the treatment of certain subsets of asthma patients. The value of FeNO really lies in repeated measures of FeNO over time due to strong sources of heterogeneity across people. The premise of population-level cutoffs using cross-sectional FeNO data is weak, and this article adds little value to the literature. Even for the goal of demonstrating biostatistical methodology, the authors use a definition of accuracy which is a function of prevalence, and then remark that accuracy is a function of prevalence. The authors also conclude that the best accuracy of FeNO was to identify questionnaire report of asthma in an epidemiological study. In an epidemiological study, it would be much easier to simply ask the participant if they have asthma rather than to measure FeNO. The authors miss the point of FeNO assessment in epidemiological studies, where it can certainly be of great value – but not in detecting asthma as presented here.

Author Response

Dear Reviewer,

thank you for your time and effort that you put for improving the manuscript.

Please, kindly see the attached responses.

Kind regards,

Round 2

Reviewer 1 Report

Please use spell check in word. I am not going to do this for you next time.

Abstract line 30: Incorrect spelling: respectievely should be respectively

Line 55: Know should be changed to known in this sentence: "It is less know how FeNO measurement acts in epidemiological conditions."

Line 73: Check the spelling: measuremnt should be measurement

Line 150: i is missing in rhinits